# The Potential Impact of Antibiotic Exposure on the Microbiome and Human Health

**DOI:** 10.3390/microorganisms13030602

**Published:** 2025-03-05

**Authors:** Siqi Li, Jiahao Liu, Xinyang Zhang, Qihong Gu, Yutong Wu, Xiaobo Tao, Tian Tian, Gongbu Pan, Minjie Chu

**Affiliations:** 1Department of Epidemiology, School of Public Health, Nantong University, 9 Seyuan Road, Nantong 226019, China; 2317320009@stmail.ntu.edu.cn (S.L.); 15862703103@163.com (J.L.); 2417310025@stmail.ntu.edu.cn (Q.G.); stoppeddeer@icloud.com (Y.W.); txb791471146@163.com (X.T.); ttyes_01@163.com (T.T.); 2School of Medical, Nantong University, Nantong 226019, China; 2331320092@stmail.ntu.edu.cn; 3Wicking Dementia Research and Education Centre, University of Tasmania, Hobart, TAS 7005, Australia

**Keywords:** antibiotics, antibiotic resistance, microbiota, population, novel probiotics

## Abstract

Antibiotics are a cornerstone of modern medicine, saving countless lives. However, their widespread use presents two major challenges. First, antibiotic-induced changes in the microbiome can disrupt immune function, increasing the susceptibility to diseases associated with these alterations. Second, prolonged antibiotic use fosters the proliferation of antibiotic resistance genes, leading to the emergence of resistant strains and threatening our ability to control infections. These challenges highlight an urgent global health crisis, necessitating in-depth investigation into the multifaceted effects of antibiotic exposure on microbiome dynamics and human health. In this review, we explore the potential effects of antibiotic exposure on the microbiome and its implications for overall health. Additionally, we examine the role of emerging technologies in addressing these challenges and in shaping future antibiotic development. Our goal is to provide insights that will inform more effective public health strategies and interventions aimed at mitigating the adverse consequences of antibiotic use, restoring microbial balance, and improving overall health outcomes.

## 1. Introduction

During a person’s lifetime, various microorganisms are encountered, and some of these microorganisms will colonize the human body. In fact, the human body is already inhabited by a diverse range of archaea, bacteria, fungi, viruses, and microeukaryotes prior to birth [1,2]. These microorganisms support the host by modulating the immune system, extracting nutrients for energy, maintaining ecological balance, and defending against pathogens [3]. The dynamic interactions between these microorganisms—whether beneficial or harmful—and their hosts are closely linked to overall health and the development of diseases. Notably, the impact of microorganisms on disease susceptibility and outcomes is particularly significant. Diseases can arise when foreign invaders enter the body, grow, and persist, disrupting the host’s immune equilibrium and causing cellular and tissue damage [4,5,6,7].

In the era before the advent of antibiotics, infections were a leading factor in deaths, especially pneumonia and influenza [8]. However, the discovery of penicillin by Alexander Fleming revolutionized medicine, saving countless lives from life-threatening bacterial infections and transforming the treatment of infectious diseases [9,10]. As a result, antibiotics are widely recognized as one of the greatest public health achievements of the 20th century, contributing significantly to increased life expectancy and improved child survival rates worldwide [11,12]. Furthermore, antibiotics have also prompted a shift in the spectrum of disease, bringing modern medicine into a new era. For example, in the United States, the leading cause of death has transitioned from communicable diseases to non-communicable diseases, significantly extending human lifespan [10]. Undoubtedly, there is no doubt that antibiotics have greatly improved modern lifestyles, whether in developed countries or in developing countries. This impact arises from antibiotics enabling physicians to perform intricate and delicate medical procedures (e.g., tumor resection, chemotherapy, and cesarean sections) that extend patients’ lives and improve patients’ quality of life, achieved through effective infection management and control. Beyond their direct benefits to individual health, antibiotics also play a critical role in public health. Their efficacy in inhibiting disease transmission among individuals has provided substantial benefits in preventing the spread of infectious diseases. However, the widespread use of antibiotics has also been associated with significant alterations in the human microbiome, leading to long-term health consequences

Unfortunately, the overuse of antibiotics and the resulting selection pressure on microbes have led to an unforeseen consequence: the exacerbation of antimicrobial resistance (AMR). This has gradually reduced the effectiveness of antibiotics against infections over time [13]. Moreover, a growing number of bacteria are developing resistance to multiple antibiotics, resulting in the emergence of multi-drug resistant (MDR) bacteria [14]. These bacteria have placed an increasingly heavy burden on public health systems worldwide. On the one hand, high morbidity and mortality in clinical settings are closely associated with greater challenges in infection control, with them acting as key drivers of this reality [15]. A systematic analysis assessed the deaths and disability-adjusted life years (DALYs) associated with AMR in 204 countries and territories in 2019. The study reported that approximately 4.95 million deaths were indirectly linked to AMR, with 1.27 million deaths directly attributed to AMR in 2019 alone [16] (Figure 1) [17]. The emergence of AMR not only compromises the efficacy of antibiotics but also disrupts the delicate balance of the microbiome, promoting the proliferation of resistant strains and further exacerbating health risks. AMR arises through mechanisms such as genetic mutations, horizontal gene transfer, and the selective pressure exerted by antibiotic use, enabling bacteria to survive and proliferate despite treatment.

On the other hand, apart from death and disability, prolonged illness results in lengthier hospital stays, the need for more expensive medications, and a heavier financial burden for those affected. By 2050, global economic losses attributable to AMR are projected to range between USD 300 billion and USD 1 trillion [18]. AMR is a natural phenomenon driven by evolution and selection pressures, existing independently of human misuse. While improper antibiotic use accelerates AMR, microorganisms inherently possess antibiotic resistance genes (ARGs), which are often found in resistant pathogens [19]. AMR enrichment typically begins in the patient’s gut microbiome. Antibiotics not only kill susceptible bacteria but also enrich resistant strains by selecting for those carrying ARGs. This process is particularly pronounced in the gut microbiome, where the abundance of ARGs can increase dramatically following antibiotic exposure [20,21]. Ultimately, AMR is a challenge we will all face—it is only a matter of time.

The development of new antibiotics is widely considered a potential solution to the AMR crisis. According to a World Health Organization (WHO) report in 2024, the current clinical antimicrobial pipeline contains 97 antimicrobial agents and/or combinations. However, this is still insufficient to address the growing threat of drug-resistant infections [22]. Although antibiotics were developed for human and animal pathogens, their effect targets (e.g., cell walls, ribosomes, and RNA polymerases) are not unique (Figure 2, Table 1). This allows antibiotics to indiscriminately affect both pathogenic and benign bacteria, thereby disrupting the balance of numerous metabolic transformations. In this review, we discuss how antibiotics acutely and chronically influence the complexity of microbial communities. We also elucidate the relevant factors contributing to clinical scenarios where antimicrobial treatment fails and propose potential strategies to mitigate these challenges (Figure 3).

## 2. Effects of Antibiotics on Microbial Composition in Different Populations

### 2.1. Effects of Antibiotics Use During Pregnancy and Lactation

Infections associated with pregnancy and childbirth pose a significant threat to maternal health [23]. As a result, mothers are frequently administered prophylactic antibiotics alongside standard therapeutic antibiotics to prevent and treat infections in current perinatal care. This widespread use of antibiotics during pregnancy and delivery is substantial, with reported usage rates of 37% and 33%, respectively [24]. In fact, antibiotics account for up to 80% of all prescribed medications during pregnancy [25]. Such extensive antibiotic exposure raises concerns about its potential effects on maternal and infant health, particularly in the context of microbiome development. Antibiotic exposure during early life can significantly impact the microbiome, limiting its diversity and hindering maturation, which may disrupt immune tolerance to key symbiotic microorganisms. For example, the use of antibiotics during the prenatal and perinatal periods is associated with changes in the beta diversity of the infant meconium microbiome [26]. Another study showed that antibiotic exposure reduced the load of *Lactobacillus* spp. in the maternal vaginal and neonatal meconium microbiota. Additionally, antibiotic exposure is associated with increased microbial composition differences and decreased dissimilarity of microbial structures [27]. These findings highlight the potential for antibiotic exposure to alter the early-life microbiome in ways that may have long-term health implications.

To better understand the mechanisms underlying these effects, researchers have turned to animal models. Using the interleukin (IL)-10-deficient mouse model of colitis, Miyoshi et al. investigated the impact of perinatal antibiotic administration (cefoperazone) on the microbiota of mothers and their offspring [28]. The results demonstrated that offspring from antibiotic-exposed mothers exhibit persistent gut dysbiosis and loss of immune tolerance in adulthood, thereby increasing the risk of developing inflammatory bowel disease (IBD). Similarly, Schulfer et al. also pointed out that pregnant dams inoculated with gut microbiota communities shaped by antibiotic exposure can accurately transmit their disrupted microbiomes to their offspring through vertical transmission, influencing the immune development of the offspring [29]. Yoon et al. further demonstrated that altered levels of pancreatic protease activity induced by antibiotic therapy may lead to both acute and long-term effects [30]. In the short term, increased pancreatic protease activity may enhance intestinal permeability. However, sustained elevations can compromise intestinal barrier integrity and exacerbate intestinal inflammation. These animal studies provide valuable insights into the potential mechanisms by which maternal antibiotic exposure may affect the offspring’s health.

In human studies, maternal antibiotic consumption during pregnancy has been consistently linked to changes in the microbiome composition of infants [31,32]. A prospective study indicates that alterations in early-life gut microbiota may disrupt immune development and increase the lifetime risk of immune-mediated diseases [33]. Moreover, a cohort study indicates that antibiotic use during pregnancy is associated with an increased risk of asthma in offspring, and there is a dose–response relationship between the frequency of maternal antibiotic treatment and childhood asthma risk [34]. In addition, similar to other toxins [35,36], antibiotic intake during pregnancy has been linked to multiple adverse outcomes in offspring. These include functional impairments (e.g., developmental and cognitive deficits), central nervous system disorders [37], obesity [38], immune system dysfunction [39], and diabetes [40].

The clinical use of antibiotics during pregnancy, particularly intrapartum antibiotic prophylaxis (IAP), is guided by the need to prevent serious infections. According to the American College of Obstetricians and Gynecologists (ACOG) guidelines, IAP during delivery is an effective strategy to prevent vertical transmission of Group B Streptococcus (GBS) to newborns [41]. This improves our understanding of the impact of maternal antibiotic treatment on the infant microbiome. A longitudinal prospective cohort study compared the microbial community differences among three groups: vaginally born infants not exposed to antibiotics, infants exposed to IAP for GBS, and infants delivered via cesarean section (also exposed to IAP). The study used 16S rRNA gene analysis [31]. The results indicated significant differences in the gut microbiomes between vaginally born infants not exposed to IAP and those exposed to IAP, including both GBS and cesarean section infants. Specifically, infants exposed to IAP exhibited delayed colonization of *Actinobacteria*. Many studies have shown that infants born to mothers who received IAP exhibit a decreased relative abundance of *Actinobacteria* and *Bacteroidetes* [42,43], increased proportions of *Proteobacteria* and *Firmicutes* [42], more abundant families of *Proteobacteria* [44], and reduced abundance of *Bifidobacteria* [45]. Other studies have drawn similar associations between maternal IAP exposure and alterations in infant microbiota. A study compared the initial oral microbiota of neonates born to mothers without antibiotic exposure during delivery to those born to mothers who received antibiotic prophylaxis [46]. The results indicated that newborns born to mothers with antibiotics intake exhibited a significant enrichment of phyla *Actinobacteria*, *Bacteroidetes*, and *Proteobacteria*. In contrast, the genus *Lactobacillus* (belonging to the *Firmicutes*) was more abundant in newborns without antibiotic exposure. Another study reported that maternal exposure to IAP is associated with a reduction in the absolute quantities of *Bifidobacteria* in vaginally delivered infants. It also found a decreased relative abundance of *Bifidobacteria* compared to other gut microbiota [47].

In summary, maternal antibiotic treatment is a key regulatory factor in the establishment of the early microbiota in newborns. Maternal exposure to antibiotics can significantly affect the composition and diversity of the neonatal microbiota, potentially leading to changes in the absolute numbers and relative abundances of certain bacteria within the neonatal gut microbiome, inhibiting the growth of beneficial microorganisms and consequently impacting the development of the newborn’s immune system and overall health.

### 2.2. Effects of Antibiotics on Infants

Antibiotics are the most commonly prescribed medications in the neonatal intensive care unit (NICU) due to various health conditions [48]. However, antibiotic exposure in infants can have profound effects on their gut microbiome, potentially leading to long-term health consequences. Antibiotic exposure in infants may alter the structure of their gut microbiome, leading to a reduction in the relative abundance of beneficial microorganisms and an increase in the levels of antibiotic resistance genes [49,50]. These effects are supported by a recent study involving 58 infants, which demonstrated that antibiotic exposure not only reduces the overall diversity of gut bacteria but also promotes the growth of pathogenic *Enterobacteriaceae*, decreases the abundance of beneficial organisms, and enhances the persistence of ARG-carrying bacteria [51]. The duration of antibiotic exposure plays a critical role in shaping the infant gut microbiome. A study by Zwittink et al. indicated that both short-term and long-term antibiotic treatments significantly reduce the abundance of *Bifidobacterium* [52]. Furthermore, long-term treatment results in a sustained decrease in *Bifidobacterium* abundance, which persists until the sixth week postnatally. A study involving 28 infants noted that antibiotic-treated infants showed a significant reduction in *Bacteroidetes* and *Bifidobacteria* compared to infants without antibiotic exposure [53]. These findings are further supported by a comprehensive analysis of 9 randomized controlled trials and 38 observational studies, which revealed that prolonged antibiotic exposure in infants is associated with an increased risk of necrotizing enterocolitis and/or death [54]. Additionally, many studies have demonstrated that early and prolonged use of antibiotics also increases the risk of conditions such as late-onset sepsis, bronchopulmonary dysplasia, and retinopathy of prematurity [55,56,57]. Furthermore, Yu et al. pointed out that early antibiotic use was associated with increased antibiotic use later and with higher rates of prolonged antibiotic exposure in infants [56]. This may, in turn, pose greater health risks to infants, as studies have reported that the duration of antibiotic intake is correlated with an increased risk of mortality or more severe disease conditions in this population [56,57,58]. A randomized, double-blind, placebo-controlled trial involving 59 infants aged 12–36 months to explore the effects of azithromycin on the infant gut microbiota was conducted. The results showed that azithromycin use resulted in a 23% reduction in observed abundance and a 13% reduction in Shannon diversity [59]. Similar conclusions regarding the impact of antibiotic exposure on the microbiome richness in infants have also been confirmed in other studies [50,60]. Associations between antibiotic exposure and microbial species diversity have been reported. A cohort study conducted in the United States, involving 43 infants, described the association between early-life microbial development and antibiotic exposure. The authors noted a significant difference in β-diversity between infants not exposed to antibiotics and those previously exposed to antibiotics [61]. Another randomized controlled trial involving 80 infants noted a reduction in alpha diversity and a decrease in Simpson’s cluster-level diversity in antibiotic treated infants compared to those without antibiotic therapy. However, beta diversity did not show a statistical difference [62].

Exposure to antibiotics in early life alters the composition of the microbial community and the quantity of beneficial bacteria. These changes may have enduring effects on immune development and metabolic capacity in infants [63,64]. Early exposure to antibiotics in infants is also associated with an increased risk of developing asthma, childhood allergic diseases, and subsequent obesity [65,66,67,68]. In a cohort study of 152,622 children, Donovan et al. found that antibiotic use in infancy significantly increased the risk of childhood asthma in a dose-dependent manner [69]. A birth cohort study of 789 children by Lee et al. in South Korea, found that prenatal antibiotic exposure was associated with childhood asthma. In addition, certain types of delivery, such as cesarean section, increase antibiotic exposure, and the increased number of exposures is associated with an increased risk of morbidity. The study also found that infants exposed to prenatal antibiotics had significantly lower α-diversity and reduced gut microbiota richness (Shannon index) at 6 months of age compared to unexposed infants. Additionally, the relative abundance of *Clostridium* was increased in exposed infants [70].

The relationship between antibiotic exposure and obesity has been explored in both animal models and human studies, though findings have been inconsistent [71,72]. Two studies using prescription records to assess antibiotic exposure in New Zealand are noteworthy. A prospective cohort study of 5128 children by Chelimo et al. and a retrospective cross-sectional study of 132,852 mothers and 151,359 children by Leong et al. both found that prenatal and early childhood antibiotic exposure was independently associated with obesity at 4 to 5 years of age in a dose-dependent manner [73,74]. However, Leong et al.’s fixed-effect analysis of siblings and twins showed no association between antibiotic exposure and obesity, suggesting that antibiotics are not an absolute factor in obesity but may still play a contributing role.

In summary, early antibiotic exposure in infants can disrupt the balance of gastrointestinal microbiota, leading to dysbiosis. This alteration in the microbial community can negatively impact the development of the immune system, potentially resulting in long-term consequences for the infant’s health and susceptibility to infections and allergic conditions. The disruption of beneficial bacteria and the overgrowth of pathogenic organisms may hinder the immune system’s ability to respond effectively to environmental challenges, further complicating the infant’s overall health trajectory.

### 2.3. Effects of Antibiotics on Adults

Many studies have examined the effects of antibiotics on microbial composition by administering them to healthy individuals. In a study by Raymond et al., 24 subjects (including 18 exposed to antibiotics for one week) were examined. The results showed that antibiotics altered microbiota composition and increased both the load and diversity of antibiotic resistance genes [75]. Notably, changes in microbial diversity persisted for up to 12 weeks after treatment, highlighting the lasting impact of antibiotic exposure. Other studies have supported these findings. For example, research has shown that antibiotic exposure alters microbial composition and promotes the growth of drug-resistant bacteria [76,77]. A study selected 12 healthy male participants to undergo a 4-day antibiotic intervention [78]. After treatment, pathogenic organisms like *Enterococcus* and *Fusobacterium* proliferated significantly, while *Bifidobacterium* species and butyrate-producing bacteria declined. Additionally, species carrying β-lactam resistance genes were positively selected.

The long-term effects of antibiotic exposure on gut microbiota have also been explored. A study analyzed stool samples from 30 healthy individuals after 10 days of antibiotic treatment [79]. In the ciprofloxacin group, *Escherichia coli* and *Bifidobacterium* levels dropped significantly by day 11. While E. coli recovered by the second month, *Enterococcus* levels remained stable for the first month, increased in the second month, and gradually returned to normal by the twelfth month. In the clindamycin group, *Lactobacillus* and *Enterococcus* levels also declined immediately after treatment. *Bifidobacterium* levels decreased sharply and did not recover until the twelfth month. These findings suggest that while the gut microbiota of healthy individuals shows resilience to short-term antibiotic exposure, such interventions leave a subtle but lasting impact. The increased burden of antibiotic resistance genes may also hinder recovery.

Antibiotic-induced disruptions in gut microbiota diversity have been linked to being overweight or obese [80]. A Japanese study found that obese individuals had significantly fewer *Bacteroidetes* and a higher *Firmicutes*-to-*Bacteroidetes* ratio compared to non-obese individuals [81]. A Ukrainian study showed that obese adults had significantly higher levels of *Firmicutes* and lower levels of *Bacteroidetes* compared to normal weight and lean individuals. This association persisted even after adjusting for confounding factors [82]. Although there is still no consistent conclusion on the microbial components associated with obesity, the studies have reported an important role for changes in functional microbial levels. Current evidence indicates that gut microbiota influences fat accumulation, insulin resistance, and low-grade inflammation, which are linked to obesity and diabetes [83]. Another study found a correlation between alterations in the microbial composition, such as reduced levels of butyrate-producing bacteria, and type 2 diabetes [84]. Antibiotics can directly affect host metabolism by disrupting microbial fermentation in the gut. For instance, the reduced abundance of butyrate-producing bacteria (e.g., *Faecalibacterium prausnitzii*) following antibiotic exposure results in decreased butyrate levels. Butyrate regulates inflammation, strengthens gut barrier integrity, and improves insulin sensitivity. As a result, antibiotic-induced changes in the microbiome can increase adiposity and insulin resistance. For example, in a mouse model, antibiotic treatment caused obesity and metabolic dysregulation [85]. Furthermore, patients with diabetes and obesity often exhibit a reduced diversity of gut microbiota, and studies suggest that long-term antibiotic exposure in early life can predispose individuals to these metabolic disorders by altering gut microbial composition and function.

Antibiotic exposure can significantly alter the structure and function of the gut microbiome in adults, potentially affecting host metabolism, immune response, and inflammatory status. Research has indicated that the use of antibiotics is associated with the development of various diseases, including obesity, diabetes, and intestinal inflammation. These changes may worsen host health by reducing gut microbiota diversity and abundance (Table 1, Figure 3). The evidence underscores the importance of understanding the long-term consequences of antibiotic use and the need for strategies to mitigate their impact on gut health.

## 3. Discussion

The growing recognition of the gut microbiome’s role in human health has made it crucial to study how antibiotics impact the microbiome and their potential risks to host health. In this study, we address the negative effects of antibiotics on human health, covering different stages from pregnancy to adulthood (Figure 4).

A deeper understanding of how antibiotics affect the human microbiome is essential to maximize their therapeutic benefits, minimize harm to gut microbiota, and reduce the risk of antimicrobial resistance transmission [13,86]. As outlined in this review, the response of the microbiome to antibiotics is highly complex and variable [87]. This complexity arises from the interactions among various bacteria, fungi, and other microorganisms within the microbiome, as well as their dynamic relationships with the host. Different antibiotic classes have distinct effects on the microbiome. Some significantly reduce specific bacterial populations, while others selectively promote the growth of resistant microorganisms. Penicillins, for example, predominantly target Gram-positive bacteria by inhibiting peptidoglycan synthesis. This results in a marked reduction in species like *Firmicutes* and Bacteroides, which are essential for nutrient absorption and immune regulation. In contrast, beta-lactam antibiotics such as amoxicillin may unintentionally promote the growth of *Proteobacteria*—including *Escherichia coli*—which are more resilient due to their ability to form biofilms and harbor ARGs. Such disruptions can cause dysbiosis, leading to inflammation and long-term metabolic changes like insulin resistance. Antibiotics also disrupt the balance of microbial populations by directly affecting microbial metabolic pathways, including those involved in fermentation and short-chain fatty acid (SCFA) production. For instance, broad-spectrum antibiotics like clindamycin or ciprofloxacin may reduce the abundance of beneficial microbes like *Bifidobacterium* and *Lactobacillus*, which play key roles in maintaining gut homeostasis. The reduction in SCFA production (such as butyrate) can impair gut epithelial barrier function, leading to increased intestinal permeability and inflammation. Additionally, antibiotic-induced selective pressure promotes the overgrowth of resistant species like *Enterococcus* and *Clostridium* difficile, leading to opportunistic infections and antibiotic resistance. Although the microbiome exhibits a certain degree of resilience in the face of these impacts, the increase in the carriage of resistance genes enhances pathogen resistance, thereby placing individuals in increasingly complex and challenging situations.

Growing evidence highlights the urgent need for strong public health interventions to address antibiotic-induced microbial dysbiosis and resistance. Although many countries have implemented antibiotic stewardship programs to reduce unnecessary clinical prescriptions, overuse in agriculture and animal farming remain a significant issue, contributing to the rise in AMR. As such, public health strategies should not only focus on human health but also include environmental and agricultural interventions. GAP outlines key areas for action, including improving infection prevention, optimizing antibiotic use, and enhancing the surveillance of antimicrobial use. Strengthening international cooperation is essential to ensure the global and effective implementation of these policies. In 2015, the WHO, Food and Agriculture Organization (FAO), and World Organization for Animal Health (WOAH) endorsed the Global Action Plan (GAP) on Antimicrobial Resistance. This plan aims to tackle antimicrobial resistance through five key areas: raising awareness, monitoring and research, infection prevention and control, optimizing usage, and enhancing investment [88]. In response to the WHO’s GAP, China launched its National Action Plan to Control Bacterial Resistance (2016–2020) in 2016. This plan implemented comprehensive strategies to combat bacterial resistance and strengthened oversight of drug research, production, distribution, use, and environmental protection [89]. In 2022, China issued the new National Action Plan for Controlling Bacterial Resistance (2022–2025), which continues to prioritize infection prevention and controls to slow the spread of drug-resistant infections and reduce the demand for antibiotics [90]. Meanwhile, in 2014 the US government established the National Strategy for Combating Antibiotic-Resistant Bacteria (CARB Strategy) [91] to enhance public health efforts against antibiotic resistance threats. Subsequently, in October 2020, it unveiled the National Action Plan for Combating Antibiotic-Resistant Bacteria (2020–2025) [92]. On June 13, 2023, the European Council adopted a recommendation aimed at intensifying EU actions against AMR using a One Health approach [93]. This recommendation emphasizes infection prevention and control measures as well as monitoring and surveillance activities while also promoting innovation alongside responsible access to effective antimicrobials through collaboration among member states and globally. These policies have slowed the development of antibiotic resistance to varying degrees, but their focus has remained on the antibiotics themselves. However, the causes and effects of AMR lie in many aspects, including humans, terrestrial and aquatic animals, plants, food, feed, and the environment. Waste from antibiotic production encourages the emergence of new resistant bacteria. The WHO and the United Nations Environment Programme (UNEP) officially released guidance on wastewater and solid waste management for manufacturing of antibiotics on September 3, 2024 [94], with the aim of curbing the spread of environmental microbial resistance during the manufacturing process. These are the first ever WHO guidelines on antibiotic contamination in the pharmaceutical industry. It also suggests that national policy makers should pay attention to the environmental pollution caused by antibiotics to avoid the accelerated arrival of the post-antibiotic era.

Meanwhile, it is also important to pay attention to emerging tools for studying the long-term effects of antibiotics on the microbiome. Metagenomic next-generation sequencing (mNGS) is a technique for detecting novel or rare microorganisms with the advantages of unbiased sampling, high sensitivity analysis, etc. [95]. And the use of mNGS tests is less susceptible to interference from previous antibiotic exposure [96]. mNGS can not only identify pathogens but can also detect clinically relevant ARGs when necessary and further predict pathogen resistance to make a more reasonable diagnosis and treatment plan for patients. Dhariwal uses metagenomics to study the resistome development in the nasopharynx of preterm infants during early life, uncovering a dynamic environment affected by early-life antibiotics [97]. However, high cost is an important factor restricting the clinical development of mNGS [98]. Antibiotics are present in every aspect of life, and AMR in the soil and rivers can exacerbate the relationship between humans and nature through the food chain [99]. Li et al. developed single-cell Raman spectroscopy with isotope labeling for the analysis of active antibiotic-resistant bacteria (ARBs) inhabiting different soils and they further combined targeted single-cell sorting with mNGS to pinpoint the most active ARBs in order to answer the question of “who is doing what and how [100]”. Zheng et al. used mNGS on water from 16 estuaries in China and detected that ARG abundance increased significantly with latitude, and that global warming would lead to a decrease in ARG [101]. Blood stream infection (BSI) is the key cause of host death caused by AMR [16]. In addition to the host metabolism being affected by serious infection, the microbial metabolism also plays a significant role [102]. Mayers et al. used iterative comparative metabolomics pipelines to gain insight into the effects of antibiotics on the host microbiome to suggest better treatment options for BSI patients [103]. Single-cell RNA-seq (scRNA-seq) is used in all aspects of microbial research, but many widespread transcriptional reactions are obscured by it. In order to compensate for this shortcoming, Ma et al. introduced a new bacterial scRNA-seq method, BacDrop [104]. BacDrop is stable and reproducible, and based on this method the researchers demonstrated the state diversity of bacteria in stable populations and heterogeneous responses in dynamic populations after antibiotic disturbance. This method has provided support for subsequent research on the emergence and resolution of antibiotic resistance, persistence, and tolerance.

Addressing the antibiotic crisis requires not only limiting antibiotic use and developing new antibiotics to combat resistant bacteria, but also minimizing their adverse effects on the human body and restoring the disrupted microbiome balance. Achieving this balance is critical to reducing the long-term impact of antibiotics on human health and microbial ecosystems. Moreover, alternative strategies such as probiotics, which have the potential to serve as substitutes or adjuncts to antibiotics, merit further attention. However, their efficacy, safety, and regulatory approval remain areas of active investigation. Antibiotic exposure is a well-documented trigger for irritable bowel syndrome (IBS), and modulating the gut microbiota has shown promise in alleviating its occurrence and progression [105]. Fecal microbiota transplantation (FMT) has emerged as a promising strategy for correcting microbiota dysregulation in diarrhea-dominant irritable bowel syndrome (IBS-D) [106]. However, its clinical application is limited by challenges such as donor screening, standardization of procedures, and potential risks of infection or long-term adverse effects. A placebo-controlled trial showed that antibiotic pretreatment significantly reduced bacterial implantation after FMT in patients with IBS-D [107]. However, since antibiotic therapy typically precedes FMT, the transplanted gut microbiota may be significantly influenced, and the specific mechanisms by which FMT improves IBS-D remain poorly understood [78]. Further studies are needed to elucidate these mechanisms and optimize the therapeutic protocol. Postbiotics and paraprobiotics are emerging as promising therapeutic agents due to evidence on their potential to reduce inflammatory diseases, bacteremia, and antibiotic resistance in young and immunocompromised patients [108]. As a potential substitute for antibiotics, they can help by improving the diversity and balance of the microbiota, reduce inflammation and oxidative stress, improve barrier function, and reduce the risk of infections [109]. Probiotics not only have a general effect on pathogens but also have a specific effect on reducing the expression of resistance genes. They affect a variety of gastrointestinal disorders, including peptic ulcers, pancreatitis, and so on [110,111]. The safety and regulatory challenges associated with bioengineered probiotics and their derivatives must be rigorously addressed to ensure consistency in their production, efficacy, and safety. Standardized guidelines and long-term monitoring are essential to mitigate potential risks and facilitate their clinical translation.

## 4. Conclusions

In conclusion, while antibiotics have revolutionized medicine and saved countless lives, their overuse and misuse have led to serious public health issues like AMR and microbiome dysbiosis. Developing new antibiotics must therefore prioritize safety, efficacy, and appropriate use. Clinicians should carefully assess patient needs when prescribing antibiotics, and public health strategies should focus on microbiome restoration, rational medication, and pollution control in antibiotic production. Addressing these challenges is essential for future health security.

Addressing antibiotic-related issues requires collaboration across microbiology, pharmacology, public health, and environmental science. Interdisciplinary research can enhance our understanding of antibiotic resistance and its health impacts, while guiding effective strategies for policymakers. Countries should implement stricter antibiotic stewardship programs, and international organizations like the WHO and FAO must coordinate efforts to reduce resistance in humans and animals. Environmental measures, such as controlling pharmaceutical waste and regulating agricultural antibiotic use, should also be integrated into policies. Developing alternative therapies like probiotics and advanced diagnostics will further support these efforts.

## Figures and Tables

**Figure 1 microorganisms-13-00602-f001:**
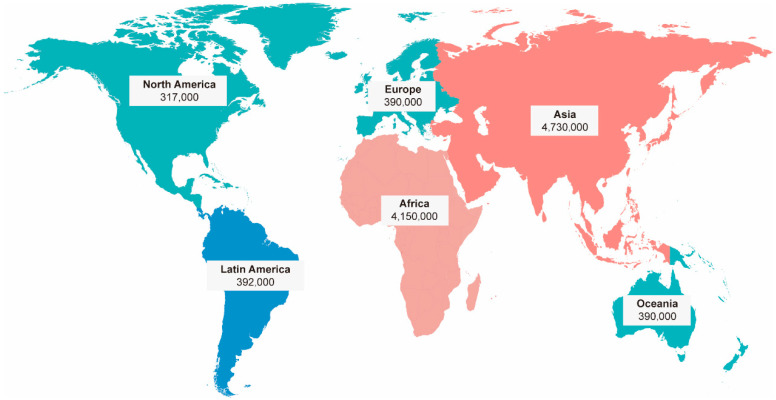
The expected annual number of deaths attributed to antimicrobial resistance in 2050 [17].

**Figure 2 microorganisms-13-00602-f002:**
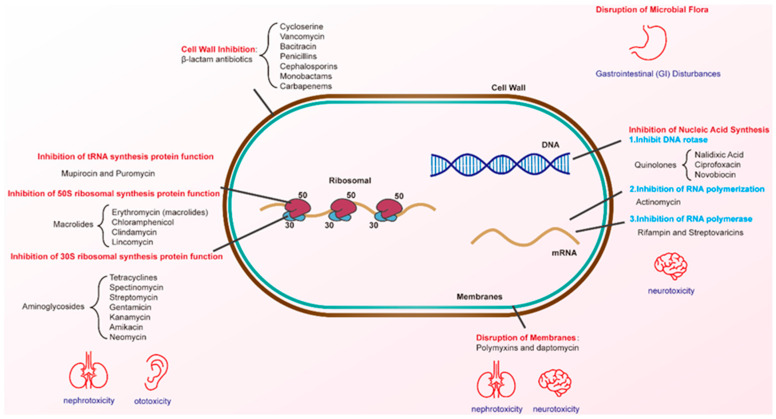
Timeline of antibiotic milestones and associated side effects. The upper blue timeline highlights key milestones in antibiotic development, while the lower red timeline outlines the associated side effects and challenges.

**Figure 3 microorganisms-13-00602-f003:**
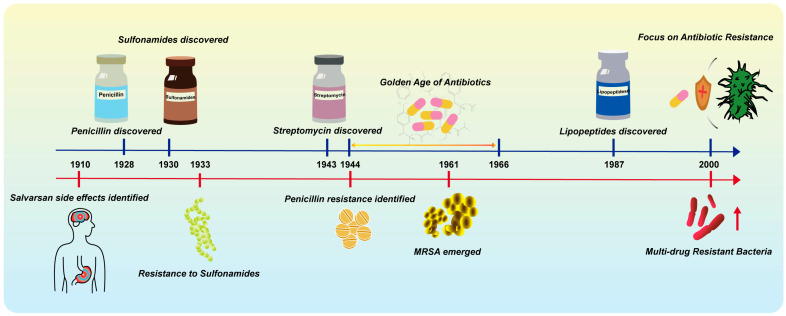
The mechanism of action of antibiotics.

**Figure 4 microorganisms-13-00602-f004:**
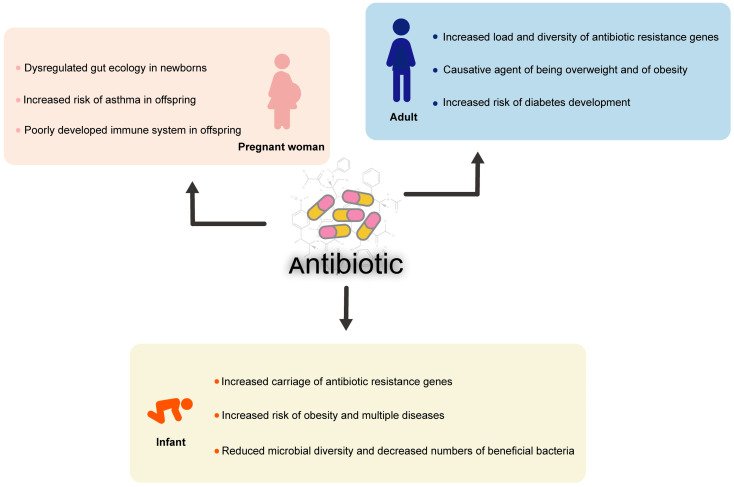
The impact of antibiotic resistance on different populations.

**Table 1 microorganisms-13-00602-t001:** Different effects of antibiotics and their representative drugs.

Mechanisms of Action	Specific Way		Examples
Inhibits cell wall synthesis		Inhibits enzyme activity and prevents cross-linking and synthesis of peptidoglycan	Penicillin and cephalosporins
Inhibits protein synthesis	50 S subu-nit	Inhibits the extension of the peptide chain	Erythromycin
	30 S subu-nit	Misinterprets mRNA, synthesizing malfunctioning proteins	Streptomycin
Inhibits nucleic acid syn-thesis		Inhibition of bacterial DNA gyrase and topoisomerase IV	Quinolone antibiotics
Interferes with metabolic pathways		Inhibition of dihydrofolate synthetase	Sulfonamides

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
