# Peer review of "The Potential Impact of Antibiotic Exposure on the Microbiome and Human Health"

_microorganisms, 2025, doi:10.3390/microorganisms13030602_

Round 1
Reviewer 1 Report
Comments and Suggestions for Authors
This review presents an important and timely discussion on the effects of antibiotic exposure on the microbiome and human health. The manuscript is well-researched, supported by relevant literature, and provides valuable insights into the challenges of antimicrobial resistance (AMR) and microbiome dysbiosis. However, there are some areas where clarity, organization, and language could be improved for better readability and impact.
Specific Suggestions
While the manuscript is well-structured, some sections contain lengthy and complex sentences that could be simplified for better comprehension. Consider breaking up long sentences and using clearer transitions between ideas.
Some redundant phrases and repetitive explanations can be streamlined to improve the flow of the text.
The discussion on the impact of antibiotics on microbial composition is comprehensive, but certain claims would benefit from additional clarification or further elaboration, especially regarding mechanistic insights into how microbiome alterations contribute to disease.
In the discussion of alternative therapies (e.g., probiotics, fecal microbiota transplantation), it may be helpful to provide a more balanced discussion of their current limitations, regulatory challenges, and ongoing research directions.
The manuscript occasionally shifts between discussing general effects and specific population groups (e.g., pregnant women, infants, adults) without clear transitions. Consider improving section transitions to enhance readability.
The conclusions could be more concise, summarizing key takeaways rather than reiterating the entire discussion.
If possible, simplify Figure 2 (Timeline of Antibiotic Milestones and Side Effects) to make it more visually engaging and less text-heavy.
Reviewer 2 Report
Comments and Suggestions for Authors
Dear authors,
Below I am sending my review statement for the publication microorganisms-3484582 titled "The potential impact of antibiotic exposure on the microbiome and human health." First of all, I must state that the issue of microorganism resistance to antibiotics is a critical problem of our time. It is also one of the biggest challenges for humanity in the coming years, even though it may not seem so at the moment. The study is primarily focused on providing a summary of information regarding the impact of antibiotics on the microbiome and human health across different age groups. In this sense, the review publication is quite unique. In my opinion, the text is beneficial and enriching.
However, I have several significant comments that must be addressed:
-
The text needs to be thoroughly reviewed and corrected for some problematic areas, inconsistent expressions, etc. For example, in most cases, there is a missing space between the word and the parenthesis with the citation. This suggests carelessness and imprecision, which is unfortunate.
-
Line 52 – “antibiotics” should have a capital “A” at the beginning.
-
Fig. 1 – citation is missing!! Who is the author, or where were the data sourced from?
-
A similar comment applies to Fig. 3 and many other objects throughout the text. Proper citation is absolutely crucial, and I believe this is a major issue in the text!
-
Table 1 – I suggest changing “Concrete way” to “Specific way”; The items in the “Examples” column should be capitalized at the beginning of each word – in line with the other cells in the table!
-
Especially in Chapter 3, there are many Latin names of microorganisms that are not correctly italicized!
-
The citations are definitely not in accordance with the current requirements of MDPI/Microorganisms (for example, the full author list is missing, Latin names are not italicized, and there is missing information in several citations, such as in citation No. 33, among others).
